# Predicting trends in the quality of state-of-the-art neural networks without access to training or testing data

Charles H. Martin[1], Tongsu (Serena) Peng[1] & Michael W. Mahoney [2✉]

In many applications, one works with neural network models trained by someone else. For such pretrained models, one may not have access to training data or test data. Moreover, one may not know details about the model, e.g., the specifics of the training data, the loss function, the hyperparameter values, etc. Given one or many pretrained models, it is a challenge to say anything about the expected performance or quality of the models. Here, we address this challenge by providing a detailed meta-analysis of hundreds of publicly available pretrained models. We examine norm-based capacity control metrics as well as power law based metrics from the recently-developed Theory of Heavy-Tailed Self Regularization. We find that norm based metrics correlate well with reported test accuracies for well-trained models, but that they often cannot distinguish well-trained versus poorly trained models. We also find that power law based metrics can do much better—quantitatively better at discriminating among series of well-trained models with a given architecture; and qualitatively better at discriminating well-trained versus poorly trained models. These methods can be used to identify when a pretrained neural network has problems that cannot be detected simply by examining training/test accuracies.

[1] Calculation Consulting, San Francisco, CA, USA. [2] ICSI and Department of Statistics, University of California at Berkeley, Berkeley, CA, USA.
✉email: mmahoney@stat.berkeley.edu

A common problem in machine learning (ML) is to evaluate the quality of a given model. A popular way to accomplish this is to train a model and then evaluate its training/testing error. There are many problems with this approach. The training/testing curves give very limited insight into the overall properties of the model; they do not take into account the (often large human and CPU/GPU) time for hyperparameter fiddling; they typically do not correlate with other properties of interest such as robustness or fairness or interpretability; and so on. A related problem, in particular in industrial-scale artificial intelligence (AI), arises when the model user is not the model developer. Then, one may not have access to the training data or the testing data. Instead, one may simply be given a model that has already been trained—a pretrained model —and need to use it as-is, or to fine-tune and/or compress it and then use it.

Naïvely—but in our experience commonly, among ML practitioners and ML theorists—if one does not have access to training or testing data, then one can say absolutely nothing about the quality of a ML model. This may be true in worst-case theory, but models are used in practice, and there is a need for a practical theory to guide that practice. Moreover, if ML is to become an industrial process, then that process will become compartmentalized in order to scale: some groups will gather data, other groups will develop models, and other groups will use those models. Users of models cannot be expected to know the precise details of how models were built, the specifics of data that were used to train the model, what was the loss function or hyperparameter values, how precisely the model was regularized, etc.

Moreover, for many large scale, practical applications, there is no obvious way to define an ideal test metric. For example, models that generate fake text or conversational chatbots may use a proxy, like perplexity, as a test metric. In the end, however, they really require human evaluation. Alternatively, models that cluster user profiles, which are widely used in areas such as marketing and advertising, are unsupervised and have no obvious labels for comparison and/or evaluation. In these and other areas, ML objectives can be poor proxies for downstream goals.

Most importantly, in industry, one faces unique practical problems such as determining whether one has enough data for a given model. Indeed, high quality, labeled data can be very expensive to acquire, and this cost can make or break a project. Methods that are developed and evaluated on any well-defined publicly available corpus of data, no matter how large or diverse or interesting, are clearly not going to be well-suited to address problems such as this. It is of great practical interest to have metrics to evaluate the quality of a trained model—in the absence of training/testing data and without any detailed knowledge of the training/testing process. There is a need for a practical theory for pretrained models which can predict how, when, and why such models can be expected to perform well or poorly.

In the absence of training and testing data, obvious quantities to examine are the weight matrices of pretrained models, e.g., properties such as norms of weight matrices and/or parameters of Power Law (PL) fits of the eigenvalues of weight matrices. Norm-based metrics have been used in traditional statistical learning theory to bound capacity and construct regularizers; and PL fits are based on statistical mechanics approaches to deep neural networks (DNNs). While we use traditional norm-based and PL-based metrics, our goals are not the traditional goals. Unlike more common ML approaches, we do not seek a bound on the generalization (e.g., by evaluating training/test errors), we do not seek a new regularizer, and we do not aim to evaluate a single model (e.g., as with hyperparameter optimization). Instead, we want to examine different models across common architecture series, and we want to compare models between different architectures themselves. In both cases, one can ask whether it is possible to predict trends in the quality of pretrained DNN models without access to training or testing data.

To answer this question, we provide a detailed empirical analysis, evaluating quality metrics for pretrained DNN models, and we do so at scale. Our approach may be viewed as a statistical meta-analysis of previously published work, where we consider a large suite of hundreds of publicly available models, mostly from computer vision (CV) and natural language processing (NLP). By now, there are many such state-of-the-art models that are publicly available, e.g., hundreds of pretrained models in CV ($\geq$500) and NLP ($\approx$100). (When we began this work in 2018, there were fewer than tens of such models; then in 2020, there are hundreds of such models; and we expect that in a year or two there will be an order of magnitude or more of such models.) For all these models, we have no access to training data or testing data, and we have no specific knowledge of the training/testing protocols. Here is a summary of our main results. First, norm-based metrics do a reasonably good job at predicting quality trends in well-trained CV/NLP models. Second, norm-based metrics may give spurious results when applied to poorly trained models (e.g., models trained without enough data, etc.). For example, they may exhibit what we call Scale Collapse for these models. Third, PL-based metrics can do much better at predicting quality trends in pretrained CV/NLP models. In particular, a weighted PL exponent (weighted by the log of the spectral norm of the corresponding layer) is quantitatively better at discriminating among a series of well-trained versus very-well-trained models within a given architecture series; and the (unweighted) average PL exponent is qualitatively better at discriminating well-trained versus poorly-trained models. Fourth, PL-based metrics can also be used to characterize fine-scale model properties, including what we call layer-wise Correlation Flow, in well-trained and poorly-trained models; and they can be used to evaluate model enhancements (e.g., distillation, fine-tuning, etc.). Our work provides a theoretically principled empirical evaluation—by far the largest, most detailed, and most comprehensive to date—and the theory we apply was developed previously[1–3]. Performing such a meta-analysis of previously published work is common in certain areas, but it is quite rare in ML, where the emphasis is on developing better training protocols.

## Results

After describing our overall approach, we study in detail three well-known CV architecture series (the VGG, ResNet, and DenseNet series of models). Then, we look in detail at several variations of a popular NLP architecture series (the OpenAI GPT and GPT2 series of models), and we present results from a broader analysis of hundreds of pretrained DNN models.

**Overall approach.** Consider the objective/optimization function (parameterized by $\mathbf{W}_l$s and $\mathbf{b}_l$s) for a DNN with $L$ layers, and weight matrices $\mathbf{W}_l$ and bias vectors $\mathbf{b}_l$, as the minimization of a general loss function $\mathcal{L}$ over the training data instances and labels, $\{\mathbf{x}_i, y_i\} \in \mathcal{D}$. For a typical supervised classification problem, the goal of training is to construct (or learn) $\mathbf{W}_l$ and $\mathbf{b}_l$ that capture correlations in the data, in the sense of solving

$$\underset{\mathbf{W}_l, \mathbf{b}_L}{\text{argmin}} \sum_{i=1}^{N} \mathcal{L}(E_{DNN}(\mathbf{x}_i), y_i), \qquad (1)$$

where the loss function $\mathcal{L}(\cdot, \cdot)$ can take on a myriad of forms[4], and where the energy (or optimization) landscape function

$$E_{DNN} = f(\mathbf{x}_i; \mathbf{W}_1, \dots, \mathbf{W}_L, \mathbf{b}_1, \dots, \mathbf{b}_L) \qquad (2)$$

depends parametrically on the weights and biases. For a trained

model, the form of the function $E_{DNN}$ does not explicitly depend on the data (but it does explicitly depend on the weights and biases). The function $E_{DNN}$ maps data instance vectors ($x_i$ values) to predictions ($y_i$ labels), and thus the output of this function does depend on the data. Therefore, one can analyze the form of $E_{DNN}$ in the absence of any training or test data.

Test accuracies have been reported online for publicly available pretrained pyTorch models[5]. These models have been trained and evaluated on labeled data $\{x_i, y_i\} \in \mathcal{D}$, using standard techniques. We do not have access to this data, and we have not trained any of the models ourselves. Our methodological approach is thus similar to a statistical meta-analysis, common in biomedical research, but uncommon in ML. Computations were performed with the publicly available WeightWatcher tool (version 0.2.7)[6]. To be fully reproducible, we only examine publicly available, pretrained models, and we provide all Jupyter and Google Colab notebooks used in an accompanying github repository[7]. See Supplementary Note 1 for details.

Our approach involves analyzing individual DNN weight matrices, for (depending on the architecture) fully connected and/or convolutional layers. Each DNN layer contains one or more layer 2D $N_l \times M_l$ weight matrices, $W_l$, or pre-activation maps, $W_{i,l}$, e.g., extracted from 2D Convolutional layers, where $N > M$. (We may drop the $i$ and/or $i, l$ subscripts below.) The best performing quality metrics depend on the norms and/or spectral properties of each weight matrix, $W$, and/or, equivalently, it's empirical correlation matrix, $X = W^T W$. To evaluate the quality of state-of-the-art DNNs, we consider the following metrics:

$$\text{Frobenius Norm} : \| W \|_F^2 = \| X \|_F = \sum_{i=1}^{M} \lambda_i \qquad (3)$$

$$\text{Spectral Norm} : \| W \|_\infty^2 = \| X \|_\infty = \lambda_{max} \qquad (4)$$

$$\text{Weighted Alpha} : \hat{\alpha} = \alpha \log \lambda_{max} \qquad (5)$$

$$\alpha-\text{Norm(or } \alpha-\text{Shatten Norm)} : \| W \|_{2\alpha}^{2\alpha} = \| X \|_\alpha^\alpha = \sum_{i=1}^{M} \lambda_i^\alpha. \qquad (6)$$

To perform diagnostics on potentially problematic DNNs, we will decompose $\hat{\alpha}$ into its two components, $\alpha$ and $\lambda_{max}$. Here, $\lambda_i$ is the $i^{th}$ eigenvalue of the $X$, $\lambda_{max}$ is the maximum eigenvalue, and $\alpha$ is the fitted PL exponent. These eigenvalues are squares of the singular values $\sigma_i$ of $W$, $\lambda_i = \sigma_i^2$. All four metrics can be computed easily from DNN weight matrices. The first two metrics are well-known in ML. The last two metrics deserve special mention, as they depend on an empirical parameter $\alpha$ that is the PL exponent that arises in the recently developed Heavy Tailed Self Regularization (HT-SR) Theory[1-3].

In the HT-SR Theory, one analyzes the eigenvalue spectrum, i.e., the Empirical Spectral Density (ESD), of the associated correlation matrices[1-3]. From this, one characterizes the amount and form of correlation, and therefore implicit self-regularization, present in the DNN's weight matrices. For each layer weight matrix $W$, of size $N \times M$, construct the associated $M \times M$ (uncentered) correlation matrix $X$. Dropping the $L$ and $l, i$ indices, one has

$$X = \frac{1}{N} W^T W.$$

If we compute the eigenvalue spectrum of $X$, i.e., $\lambda_i$ such that $Xv_i = \lambda_i v_i$, then the ESD of eigenvalues, $\rho(\lambda)$, is just a histogram of the eigenvalues, formally written as $\rho(\lambda) = \sum_{i=1}^{M} \delta(\lambda - \lambda_i)$. Using HT-SR Theory, one characterizes the correlations in a weight matrix by examining its ESD, $\rho(\lambda)$. It can be well-fit to a truncated PL

distribution, given as

$$\rho(\lambda) \sim \lambda^{-\alpha}, \qquad (7)$$

which is (at least) valid within a bounded range of eigenvalues $\lambda \in [\lambda^{min}, \lambda^{max}]$.

The original work on HT-SR Theory considered a small number of NNs, including AlexNet and InceptionV3. It showed that for nearly every $W$, the (bulk and tail) of the ESDs can be fit to a truncated PL, and that PL exponents $\alpha$ nearly all lie within the range $\alpha \in (1.5, 5)$[1-3]. As for the mechanism responsible for these properties, statistical physics offers several possibilities[8,9], e.g., self-organized criticality[10,11] or multiplicative noise in the stochastic optimization algorithms used to train these models[12,13]. Alternatively, related techniques have been used to analyze correlations and information propagation in actual spiking neurons[14,15]. Our meta-analysis does not require knowledge of mechanisms; and it is not even clear that one mechanism is responsible for every case. Crucially, HT-SR Theory predicts that smaller values of $\alpha$ should correspond to models with better correlation over multiple size scales and thus to better models. The notion of "size scale" is well-defined in physical systems, to which this style of analysis is usually applied, but it is less well-defined in CV and NLP applications. Informally, it would correspond to pixel groups that are at a greater distance in some metric, or between sentence parts that are at a greater distance in text. Relatedly, previous work observed that smaller exponents $\alpha$ correspond to more implicit self-regularization and better generalization, and that we expect a linear correlation between $\hat{\alpha}$ and model quality[1-3].

For norm-based metrics, we use the average of the log norm, to the appropriate power. Informally, this amounts to assuming that the layer weight matrices are statistically independent, in which case we can estimate the model complexity $\mathcal{C}$, or test accuracy, with a standard Product Norm (which resembles a data dependent VC complexity),

$$\mathcal{C} \sim \| W_1 \| \times \| W_2 \| \times \cdots \times \| W_L \|, \qquad (8)$$

where $\| \cdot \|$ is a matrix norm. The log complexity,

$$\log \mathcal{C} \sim \log \| W_1 \| + \log \| W_2 \| + \cdots + \log \| W_L \| = \sum_l \log \| W_l \|, \qquad (9)$$

takes the form of an average Log Norm. For the Frobenius Norm metric and Spectral Norm metric, we can use Eq. (9) directly (since, when taking log $\| W_l \|_F^2$, the 2 comes down and out of the sum, and thus ignoring it only changes the metric by a constant factor).

The Weighted Alpha metric is an average of $\alpha_l$ over all layers $l \in \{1, ..., l\}$, weighted by the size, or scale, or each matrix,

$$\hat{\alpha} = \frac{1}{L} \sum_l \alpha_l \log \lambda_{max,l} \approx \langle \log \| X \|_\alpha^\alpha \rangle, \qquad (10)$$

where $L$ is the total number of layer weight matrices. The Weighted Alpha metric was introduced previously[3], where it was shown to correlate well with trends in reported test accuracies of pretrained DNNs, albeit on a much smaller and more limited set of models than we consider here.

Based on this, in this paper, we introduce and evaluate the $\alpha$-Shatten Norm metric,

$$\sum_l \log \| X_l \|_{\alpha_l}^{\alpha_l} = \sum_l \alpha_l \log \| X_l \|_{\alpha_l}. \qquad (11)$$

For the $\alpha$-Shatten Norm metric, $\alpha_l$ varies from layer to layer, and so in Eq. (11) it cannot be taken out of the sum. For small $\alpha$, the Weighted Alpha metric approximates the Log $\alpha$-Shatten norm, as can be shown with a statistical mechanics and random matrix theory derivation; and the Weighted Alpha and $\alpha$-Shatten

# Analyzing DNN Weight matrices with WeightWatcher

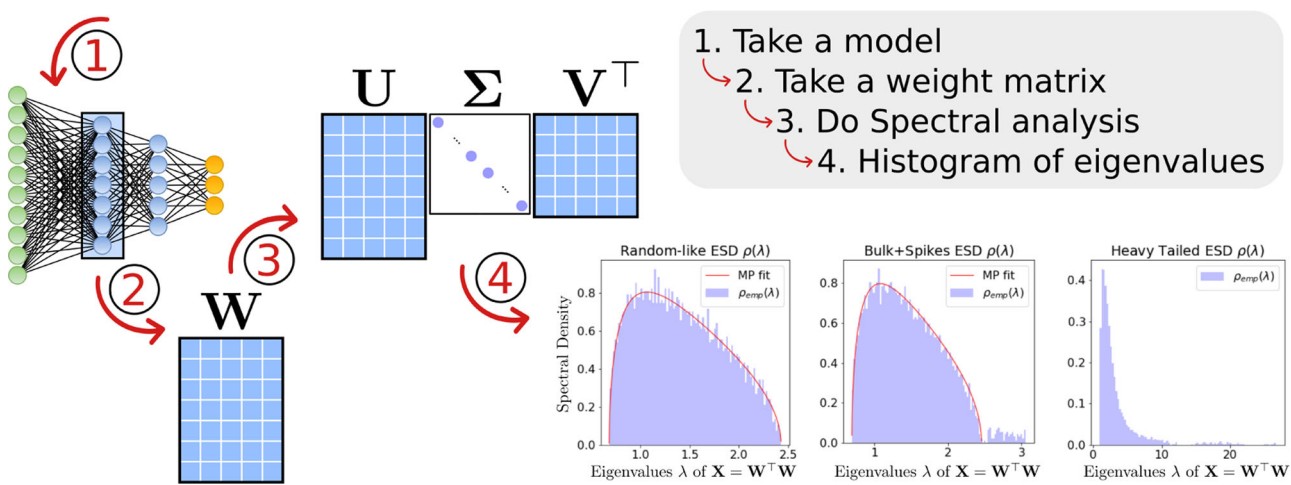

**Fig. 1 Schematic of analyzing DNN layer weight matrices W.** Given an individual layer weight matrix **W**, from either a fully connected layer or a convolutional layer, perform a Singular Value Decomposition (SVD) to obtain $\mathbf{W} = \mathbf{U\Sigma V}^T$, and examine the histogram of eigenvalues of $\mathbf{W}^T\mathbf{W}$. Norm-based metrics and PL-based metrics (that depend on fitting the histogram of eigenvalues to a truncated PL) can be used to compare models. For example, one can analyze one layer of a pre-trained model, compare multiple layers of a pre-trained model, make comparisons across model architectures, monitor neural network properties during training, etc.

norm metrics often behave like an improved, weighted average Log Spectral Norm.

Finally, although it does less well for predicting trends in state-of-the-art model series, e.g., as depth changes, the average value of $\alpha$, i.e.,

$$\bar{\alpha} = \frac{1}{L}\sum_l \alpha_l = \langle \alpha \rangle, \tag{12}$$

can be used to perform model diagnostics, to identify problems that cannot be detected by examining training/test accuracies, and to discriminate poorly trained models from well-trained models.

One determines $\alpha$ for a given layer by fitting the ESD of that layer's weight matrix to a truncated PL, using the commonly accepted Maximum Likelihood method[16,17]. This method works very well for exponents between $\alpha \in (2, 4)$; and it is adequate, although imprecise, for smaller and especially larger $\alpha$[18]. Operationally, $\alpha$ is determined by using the `WeightWatcher` tool[6] to fit the histogram of eigenvalues, $\rho(\lambda)$, to a truncated PL,

$$\rho(\lambda) \sim \lambda^\alpha, \quad \lambda \in [\lambda_{min}, \lambda_{max}], \tag{13}$$

where $\lambda_{max}$ is the largest eigenvalue of $\mathbf{X} = \mathbf{W}^T\mathbf{W}$, and where $\lambda_{min}$ is selected automatically to yield the best (in the sense of minimizing the K-S distance) PL fit. Each of these quantities is defined for a given layer **W** matrix. See Fig. 1 for an illustration.

To avoid confusion, let us clarify the relationship between $\alpha$ and $\hat{\alpha}$. We fit the ESD of the correlation matrix **X** to a truncated PL, parameterized by 2 values: the PL exponent $\alpha$, and the maximum eigenvalue $\lambda_{max}$. The PL exponent $\alpha$ measures the amount of correlation in a DNN layer weight matrix **W**. It is valid for $\lambda \le \lambda_{max}$, and it is scale-invariant, i.e., it does not depend on the normalization of **W** or **X**. The $\lambda_{max}$ is a measure of the size, or scale, of **W**. Multiplying each $\alpha$ by the corresponding $\log \lambda_{max}$ weighs "bigger" layers more, and averaging this product leads to a balanced, Weighted Alpha metric $\hat{\alpha}$ for the entire DNN. We will see that for well-trained CV and NLP models, $\hat{\alpha}$ performs quite well and as expected, but for CV and NLP models that are potentially problematic or less well-trained, metrics that depend on the scale of the problem can perform anomalously. In these

cases, separating $\hat{\alpha}$ into its two components, $\alpha$ and $\lambda_{max}$, and examining the distributions of each, can be helpful.

**Comparison of CV models**. Each of the VGG, ResNet, and DenseNet series of models consists of several pretrained DNN models, with a given base architecture, trained on the full ImageNet[19] dataset, and each is distributed with the current open source pyTorch framework (version 1.4)[20]. In addition, we examine a larger set of ResNet models, which we call the ResNet-1K series, trained on the ImageNet-1K dataset[19] and provided on the OSMR Sandbox[5]. For these models, we first perform coarse model analysis, comparing and contrasting the four model series, and predicting trends in model quality. We then perform fine layer analysis, as a function of depth. This layer analysis goes beyond predicting trends in model quality, instead illustrating that PL-based metrics can provide novel insights among the VGG, ResNet/ResNet-1K, and DenseNet architectures.

We examine the performance of the four quality metrics—Log Frobenius norm ($\langle \log \| \mathbf{W} \|_F^2 \rangle$), Log Spectral norm ($\langle \log \| \mathbf{W} \|_\infty^2 \rangle$), Weighted Alpha ($\hat{\alpha}$), and Log $\alpha$-Norm ($\langle \log \| \mathbf{X} \|_\alpha^\alpha \rangle$)—applied to each of the VGG, ResNet, ResNet-1K, and DenseNet series. Figure 2 plots the four quality metrics versus reported test accuracies[20], as well as a basic linear regression line, for the VGG series. (These test accuracies have been previously reported and made publicly available by others. We take them as given. We do not attempt to reproduce/verify them, since we do not permit ourselves access to training/test data.) Here, smaller norms and smaller values of $\hat{\alpha}$ imply better generalization (i.e., greater accuracy, lower error). Quantitatively, Log Spectral norm is the best; but, visually, all four metrics correlate quite well with reported Top1 accuracies. The DenseNet series has similar behavior. (These and many other such plots can be seen on our publicly available repo.)

To examine visually how the four quality metrics depend on data set size on a larger, more complex model series, we next look at results on ResNet versus ResNet-1K. Figure 3 compares the Log $\alpha$-Norm metric for the full ResNet model, trained on the full ImageNet dataset, against the ResNet-1K model, trained on a much smaller ImageNet-1K data set. Here, the Log $\alpha$-Norm is

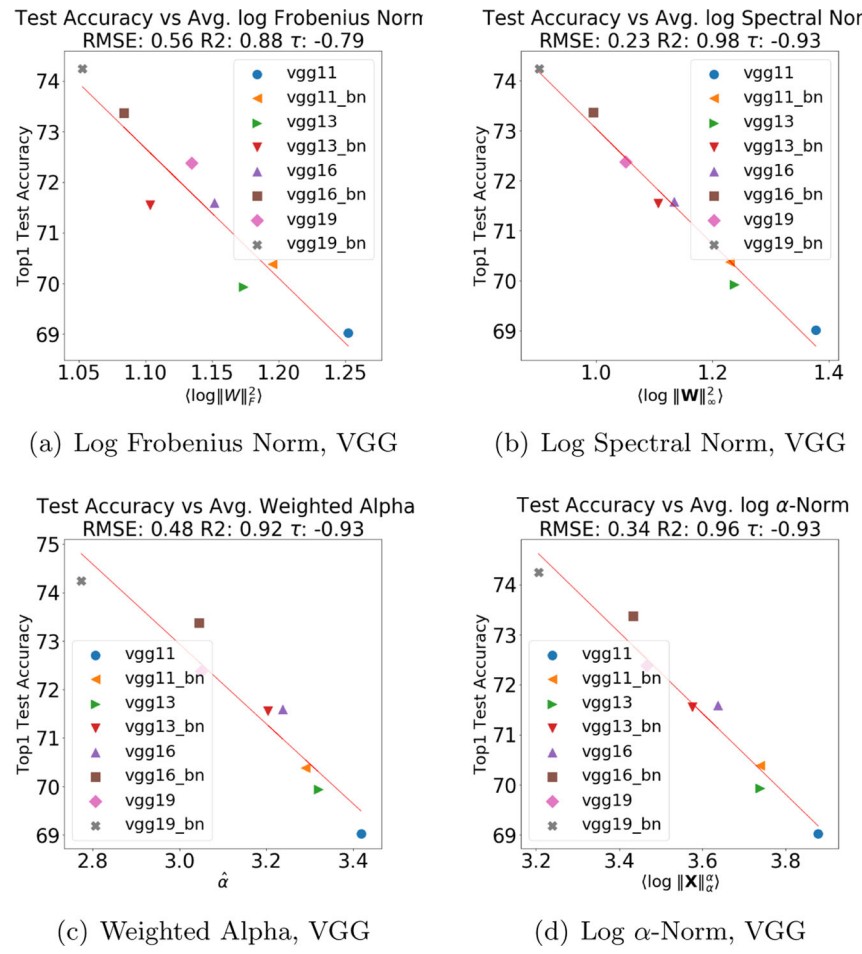

**Fig. 2 Comparison of average Log Norm and Weighted Alpha quality metrics for CV models.** Comparison of average Log Norms (in (**a**), (**b**), and (**d**)) and Weighted Alpha (in (**c**)) quality metrics versus reported test accuracy for pretrained VGG models: VGG11, VGG13, VGG16, and VGG19, with and without Batch Normalization (BN), trained on ImageNet, available in pyTorch (v1.4). Metrics fit by linear regression, RMSE, R2, and the Kendal-tau rank correlation metric reported.

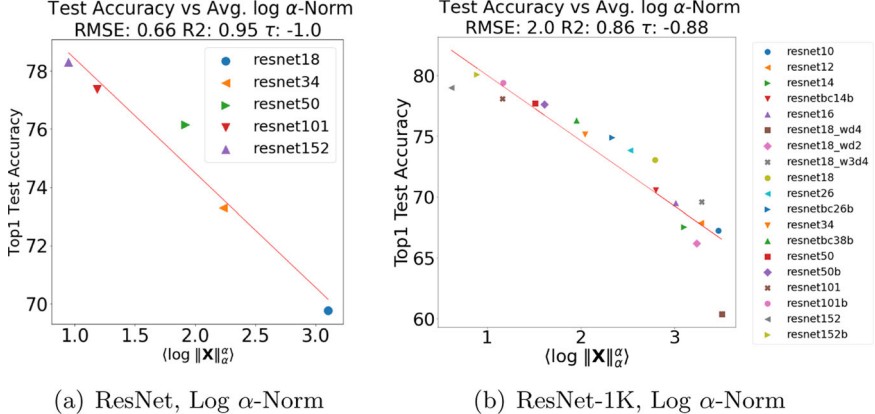

**Fig. 3 Comparison of average $\alpha$-Norm quality metric for CV models.** Comparison of average $\alpha$-Norm quality metric versus reported Top1 test accuracy for the ResNet (in (**a**)) and ResNet-1K (in (**b**)) pretrained (pyTorch) models. Metrics fit by linear regression, RMSE, R2, and the Kendal-tau rank correlation metric reported.

much better than the Log Frobenius/Spectral norm metrics (although, as Table 1 shows, it is slightly worse than the Weighted Alpha metric). The ResNet series has strong correlation (RMSE of 0.66, $R^2$ of 0.9, and Kendall-$\tau$ of $-1.0$), whereas the ResNet-1K series also shows good but weaker correlation (much larger RMSE of 1.9, $R^2$ of 0.88, and Kendall-$\tau$ of $-0.88$).

See Table 1 for a summary of results for Top1 accuracies for all four metrics for the VGG, ResNet, ResNet-1K, and DenseNet series. Similar results are obtained for the Top5 accuracies. The Log Frobenius norm performs well but not extremely well; the Log Spectral norm performs very well on smaller, simpler models like the VGG and DenseNet architectures; and, when moving to

**Table 1 Quality metrics (for RMSE, smaller is better; for $R^2$, larger is better; for Kendall-$\tau$ rank correlation, larger magnitude is better; best is bold) for reported Top1 test error for pretrained models in each architecture series.**

| Series | # | Metric | $\langle \log \| \mathbf{W} \|^2_F \rangle$ | $\langle \log \| \mathbf{W} \|^2_\infty \rangle$ | $\hat{\alpha}$ | $\langle \log \| \mathbf{X} \|^\alpha_\alpha \rangle$ |
|---|---|---|---|---|---|---|
| VGG | 6 | RMSE | 0.56 | **0.23** | 0.48 | 0.34 |
| | | $R^2$ | 0.88 | **0.98** | 0.92 | 0.96 |
| | | Kendall-$\tau$ | −0.79 | **−0.93** | **−0.93** | **−0.93** |
| ResNet | 5 | RMSE | 0.9 | 0.97 | **0.61** | 0.66 |
| | | $R^2$ | 0.92 | 0.9 | **0.96** | 0.9 |
| | | Kendall-$\tau$ | −1.0 | −1.0 | −1.0 | −1.0 |
| ResNet-1K | 19 | RMSE | 2.4 | 2.8 | **1.8** | 1.9 |
| | | $R^2$ | 0.81 | 0.74 | **0.89** | 0.88 |
| | | Kendall-$\tau$ | −0.79 | −0.79 | **−0.89** | −0.88 |
| DenseNet | 4 | RMSE | 0.3 | **0.11** | 0.16 | 0.21 |
| | | $R^2$ | 0.93 | **0.99** | 0.98 | 0.97 |
| | | Kendall-$\tau$ | −1.0 | −1.0 | −1.0 | −1.0 |

Column # refers to number of models. VGG, ResNet, and DenseNet were pretrained on ImageNet. ResNet-1K was pretrained on ImageNet-1K.

the larger, more complex ResNet series, the PL-based metrics, Weighted Alpha and the Log $\alpha$-Norm, perform the best. Overall, though, these model series are all very well-trodden; and our results indicate that norm-based metrics and PL-based metrics can both distinguish among a series of well-trained versus very-well-trained models, with PL-based metrics performing somewhat (i.e., quantitatively) better on the larger, more complex ResNet series.

In particular, the PL-based Weighted Alpha and Log $\alpha$-Norm metrics tend to perform better when there is a wider variation in the hyperparameters, going beyond just increasing the depth. In addition, sometimes the purely norm-based metrics such as the Log Spectral norm can be uncorrelated or even anti-correlated with the test accuracy, while the PL-metrrics are positively correlated. See Supplementary Note 2 for additional details.

Going beyond coarse averages to examining quality metrics for each layer weight matrix as a function of depth (or layer id), our metrics can be used to perform model diagnostics and to identify fine-scale properties in a pretrained model. Doing so involves separating $\hat{\alpha}$ into its two components, $\alpha$ and $\lambda_{max}$, and examining the distributions of each. We provide examples of this.

Figure 4 plots the PL exponent $\alpha$, as a function of depth, for each layer (first layer corresponds to data, last layer to labels) for the least accurate (shallowest) and most accurate (deepest) model in each of the VGG (no BN), ResNet, and DenseNet series. (Many more such plots are available at our repo.)

In the VGG models, Fig. 4a shows that the PL exponent $\alpha$ systematically increases as we move down the network, from data to labels, in the Conv2D layers, starting with $\alpha \lesssim 2.0$ and reaching all the way to $\alpha \sim 5.0$; and then, in the last three, large, fully connected (FC) layers, $\alpha$ stabilizes back down to $\alpha \in [2, 2.5]$. This is seen for all the VGG models (again, only the shallowest and deepest are shown), indicating that the main effect of increasing depth is to increase the range over which $\alpha$ increases, thus leading to larger $\alpha$ values in later Conv2D layers of the VGG models. This is quite different than the behavior of either the ResNet-1K models or the DenseNet models.

For the ResNet-1K models, Fig. 4b shows that $\alpha$ also increases in the last few layers (more dramatically than for VGG, observe the differing scales on the Y axes). However, as the ResNet-1K models get deeper, there is a wide range over which $\alpha$ values tend to remain small. This is seen for other models in the ResNet-1K series, but it is most pronounced for the larger ResNet-1K (152) model, where $\alpha$ remains relatively stable at $\alpha \sim 2.0$, from the earliest layers all the way until we reach close to the final layers.

For the DenseNet models, Fig. 4c shows that $\alpha$ tends to increase as the layer id increases, in particular for layers toward

the end. While this is similar to the VGG models, with the DenseNet models, $\alpha$ values increase almost immediately after the first few layers, and the variance is much larger (in particular for the earlier and middle layers, where it can range all the way to $\alpha \sim 8.0$) and much less systematic throughout the network.

Overall, Fig. 4 demonstrates that the distribution of $\alpha$ values among layers is architecture dependent, and that it can vary in a systematic way within an architecture series. This is to be expected, since some architectures enable better extraction of signal from the data. This also suggests that, while performing very well at predicting trends within an architecture series, PL-based metrics (as well as norm-based metrics) should be used with caution when comparing models with very different architectures.

Figure 4 can be understood in terms of what we will call Correlation Flow. Recall that the average Log $\alpha$-Norm metric and the Weighted Alpha metric are based on HT-SR Theory[1–3], which is in turn based on the statistical mechanics of heavy tailed and strongly correlated systems[8,21–23]. There, one expects that the weight matrices of well-trained DNNs will exhibit correlations over many size scales, as is well-known in other strongly correlated systems[8,21]. This would imply that their ESDs can be well-fit by a truncated PL, with exponents $\alpha \in [2, 4]$. Much larger values ($\alpha \gg 6$) may reflect poorer PL fits, whereas smaller values ($\alpha \sim 2$), are associated with models that generalize better.

Informally, one would expect a DNN model to perform well when it facilitates the propagation of information/features across layers. In the absence of training/test data, one might hypothesize that this flow of information leaves empirical signatures on weight matrices, and that we can quantify this by measuring the PL properties of weight matrices. In this case, smaller $\alpha$ values correspond to layers in which information correlations between data across multiple scales are better captured[1,8]. This leads to the hypothesis that small $\alpha$ values that are stable across multiple layers enable better correlation flow through the network. This is similar to recent work on the information bottleneck[24,25], except that here we work in an entirely unsupervised setting.

The similarity between norm-based metrics and PL-based metrics may lead one to wonder whether the Weighted Alpha metric is just a variation of more familiar norm-based metrics. Among hundreds of pretrained models, there are "exceptions that prove the rule", and these can be used to show that fitted $\alpha$ values do contain information not captured by norms. To illustrate this, we show that some compression/distillation methods[26] may actually damage models unexpectedly, by introducing what we call Scale Collapse, where several distilled layers have unexpectedly small Spectral Norms. By Scale Collapse, we mean that the

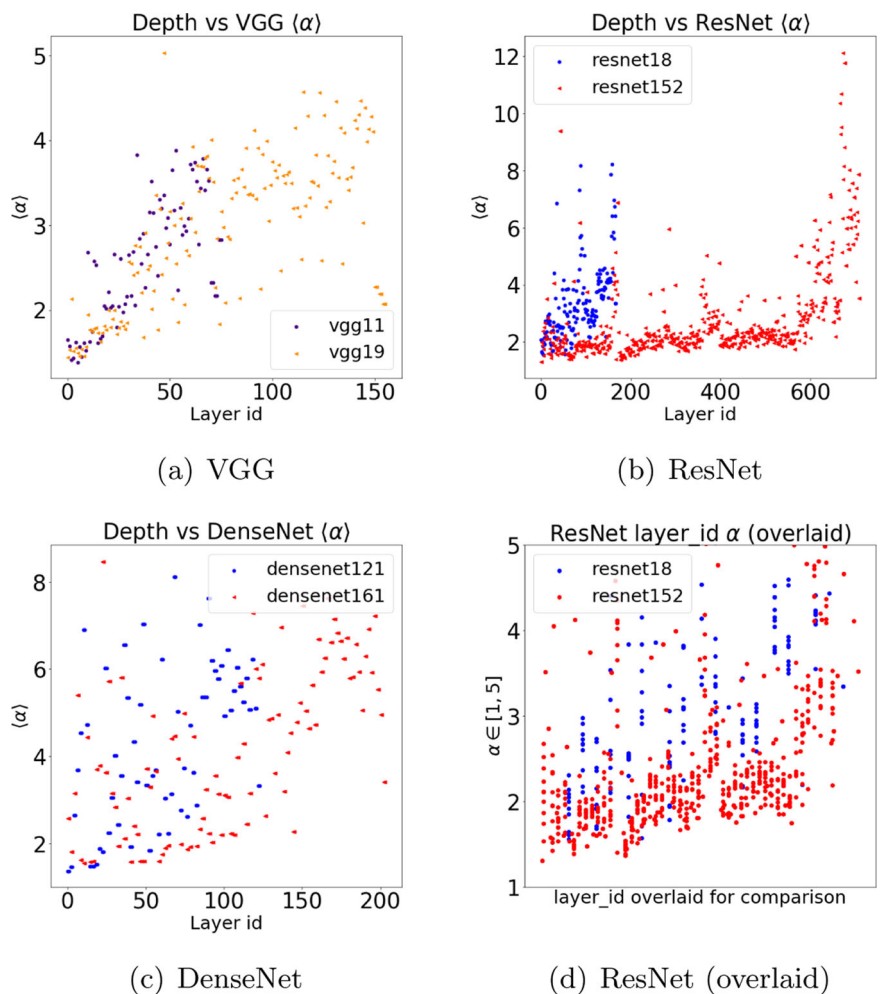

(a) VGG

(b) ResNet

(c) DenseNet

(d) ResNet (overlaid)

**Fig. 4 PL exponent ($\alpha$) versus layer id for VGG, ResNet, and DenseNet.** PL exponent ($\alpha$) versus layer id, for the least and the most accurate models in VGG (**a**), ResNet (**b**), and DenseNet (**c**) series. (VGG is without BN; and note that the Y axes on each plot are different.) Subfigure (**d**) displays the ResNet models (**b**), zoomed in to $\alpha \in [1, 5]$, and with the layer ids overlaid on the X-axis, from smallest to largest, to allow a more detailed analysis of the most strongly correlated layers. Notice that ResNet152 exhibits different and much more stable behavior of $\alpha$ across layers. This contrasts with how both VGG models gradually worsen in deeper layers and how the DenseNet models are much more erratic. In the text, this is interpreted in terms of Correlation Flow.

size scale, e.g., as measured by the Spectral or Frobenius Norm, of one or more layers changes dramatically, while the size scale of other layers changes very little, as a function of some change to or perturbation of a model. The size scales of different parts of a DNN model are typically defined implicitly by the model training process, and they typically vary in a gradual way for high-quality models. Examples of changes of interest include model compression or distillation (discussed here for a CV model), data augmentation (discussed below for an NLP model), additional training, model fine-tuning, etc.

Consider ResNet20, trained on CIFAR10, before and after applying the Group Regularization distillation technique, as implemented in the `distiller` package[27]. We analyze the pretrained 4D_regularized_5Lremoved baseline and fine-tuned models. The reported baseline test accuracies (Top1 = 91.45 and Top5 = 99.75) are better than the reported fine-tuned test accuracies (Top1 = 91.02 and Top5 = 99.67). Because the baseline accuracy is greater, the previous results on ResNet (Table 1 and Fig. 3) suggest that the baseline Spectral Norms should be smaller on average than the fine-tuned ones. The opposite is observed. Figure 5 presents the Spectral Norm (here denoted $\log \lambda_{max}$) and PL exponent ($\alpha$) for each individual layer weight

matrix **W**. On the other hand, the $\alpha$ values (in Fig. 5b) do not differ systematically between the baseline and fine-tuned models. Also, $\bar{\alpha}$, the average unweighted baseline $\alpha$, from Eq. (12), is smaller for the original model than for the fine-tuned model (as predicted by HT-SR Theory, the basis of $\hat{\alpha}$). In spite of this, Fig. 5b also depicts two very large $\alpha \gg 6$ values for the baseline, but not for the fine-tuned, model. This suggests the baseline model has at least two over-parameterized/under-trained layers, and that the distillation method does, in fact, improve the fine-tuned model by compressing these layers.

Pretrained models in the `distiller` package have passed some quality metric, but they are much less well-trodden than any of the VGG, ResNet, or DenseNet series. The obvious interpretation is that, while norms make good regularizers for a single model, there is no reason a priori to expect them correlate well with test accuracies across different models, and they may not differentiate well-trained versus poorly trained models. We do expect, however, the PL $\alpha$ to do so, because it effectively measures the amount of information correlation in the model[1–3]. This suggests that the $\alpha$ values will improve, i.e., decrease, over time, as distillation techniques continue to improve. The reason for the anomalous behavior shown in Fig. 5 is that the

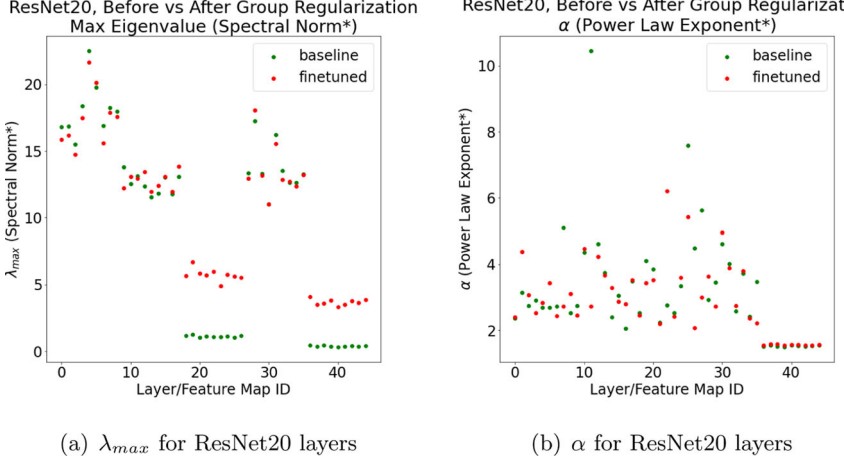

(a) $\lambda_{max}$ for ResNet20 layers

(b) $\alpha$ for ResNet20 layers

**Fig. 5** ResNet20, distilled with Group Regularization, as implemented in the `distiller` (4D_regularized_5Lremoved) pretrained models. Log Spectral Norm (log $\lambda_{max}$, in (**a**)) and PL exponent ($\alpha$, in (**b**)) for individual layers, versus layer id, for both baseline (before distillation, green) and fine-tuned (after distillation, red) pretrained models.

`distiller` Group Regularization technique causes the norms of the **W** pre-activation maps for two Conv2D layers to increase spuriously. This is difficult to diagnose by analyzing training/test curves, but it is easy to diagnose with our approach.

**Comparison of NLP Models**. Within the past few years, nearly 100 open source, pretrained NLP DNNs based on the revolutionary Transformer architecture have emerged. These include variants of BERT, Transformer-XML, GPT, etc. The Transformer architectures consist of blocks of so-called Attention layers, containing two large, Feed Forward (Linear) weight matrices[28]. In contrast to smaller pre-Activation maps arising in Cond2D layers, Attention matrices are significantly larger. In general, they have larger PL exponents $\alpha$. Based on HT-SR Theory (in particular, the interpretation of values of $\alpha \sim 2$ as modeling systems with good correlations over many size scales[8,21]), this suggests that these models fail to capture successfully many of the information correlations in the data (relative to their size) and thus are substantially under-trained. More generally, compared to CV models, modern NLP models have larger weight matrices and display different spectral properties.

While norm-based metrics perform reasonably well on well-trained NLP models, they often behave anomalously on poorly trained models. For such models, weight matrices may display rank collapse, decreased Frobenius mass, or unusually small Spectral norms. This may be misinterpreted as "smaller is better". Instead, it should probably be interpreted as being due to a similar mechanism to how distillation can "damage" otherwise good models. In contrast to norm-based metrics, PL-based metrics, including the Log $\alpha$-Norm metric and the Weighted Alpha metric, display more consistent behavior, even on less well-trained models. To help identify when architectures need repair and when more and/or better data are needed, one can use these metrics, as well as the decomposition of the Weighted Alpha metric ($\alpha \log \lambda_{max}$) into its PL component ($\alpha$) and its norm component ($\log \lambda_{max}$), for each layer.

Many NLP models, such as early variants of GPT and BERT, have weight matrices with unusually large PL exponents (e.g., $\alpha \gg 6$). This indicates these matrices may be under-correlated (i.e., over-parameterized, relative to the amount of data). In this regime, the truncated PL fit itself may not be very reliable because the Maximum Likelihood estimator it uses is unreliable in this

range. In this case, the specific $\alpha$ values returned by the truncated PL fits are less reliable, but having large versus small $\alpha$ is reliable. If the ESD is visually examined, one can usually describe these **W** as in the BULK-DECAY or BULK+SPIKES phase from HT-ST Theory[1,2]. Previous work[1,2] has conjectured that very well-trained DNNs would not have many outlier $\alpha \gg 6$. Consistent with this, more recent improved versions of GPT (shown below) and BERT (not shown) confirm this.

The OpenAI GPT and GPT2 series of models provide the opportunity to analyze two effects: increasing the sizes of both the data set and the architectures simultaneously; and training the same model with low-quality data versus high-quality data. These models have the ability to generate fake text that appears to the human to be real, and they have generated media attention because of the potential for their misuse. For this reason, the original GPT model released by OpenAI was trained on a deficient data set, rendering the model interesting but not fully functional. Later, OpenAI released a much improved model, GPT2-small, which has the same architecture and number of layers as GPT, but which has been trained on a larger and better data set, making it remarkably good at generating (near) human-quality fake text. Subsequent models in the GPT2 were larger and trained to more data. By comparing GPT2-small to GPT2-medium to GPT2-large to GPT2-xl, we can examine the effect of increasing data set and model size simultaneously, as well as analyze well-trained versus very-well-trained models. By comparing the poorly trained GPT to the well-trained GPT2-small, we can identify empirical indicators for when a model has been poorly trained and thus may perform poorly when deployed. The GPT models we analyze are deployed with the popular HuggingFace PyTorch library[29].

We examine the performance of the four quality metrics (Log Frobenius norm, Log Spectral norm, Weighted Alpha, and Log $\alpha$-Norm) for the OpenAI GPT and GPT2 pretrained models. See Table 2 for a summary of results. Comparing trends between GPT2-medium to GPT2-large to GPT2-xl, observe that (with one minor exception involving the log Frobenius norm metric) all four metrics decrease as one goes from medium to large to xl. This indicates that the larger models indeed look better than the smaller models, as expected. GPT2-small violates this general trend, but only very slightly. This could be due to under-optimization of the GPT2-small model, or since it is the smallest of the GPT2 series, and the metrics we present are most relevant

for models at scale. Aside from this minor discrepancy, overall for these well-trained models, all these metrics now behave as expected, i.e., there is no Scale Collapse and norms are decreasing with increasing accuracy.

Comparing trends between GPT and GPT2-small reveals a different story. Observe that all four metrics increase when going from GPT to GPT2-small, i.e., they are larger for the higher-quality model (higher quality since GPT2-small was trained to better data) and smaller for the lower-quality model, when the number of layers is held fixed. This is unexpected. Here, too, we can perform model diagnostics, by separating $\hat{\alpha}$ into its two components, $\alpha$ and $\lambda_{max}$ and examining the distributions of each. In doing so, we see additional examples of Scale Collapse and additional evidence for Correlation Flow.

We next examine the Spectral norm in GPT versus GPT2-small. In Fig. 6a, the poorly-trained GPT model has a smaller mean/median Spectral norm as well as, spuriously, many much smaller Spectral norms, compared to the well-trained GPT2-small. This violates the conventional wisdom that smaller Spectral norms are better. Because there are so many anomalously small Spectral norms, the GPT model appears to be exhibiting a kind of Scale Collapse, like that observed (in Fig. 5) for the distilled CV models. This demonstrates that, while the Spectral (or Frobenius) norm may correlate well with predicted test error, at least among reasonably well-trained models, it is not a good indicator of the overall model quality in general. Naïvely using it as an empirical quality metric may give spurious results when applied to poorly trained or otherwise deficient models.

Figure 7a shows the Spectral norm as a function of depth (layer id). This illustrates two phenomenon. First, the large value of Spectral norm (in Fig. 6a) corresponds to the first embedding layer(s). These layers have a different effective normalization, and therefore a different scale. See Supplementary Note 2 for details. We do not include them in our computed average metrics in Table 2. Second, for GPT, there seems to be two types of layers with very different Spectral norms (an effect which is seen, but to a much weaker extent, for GPT2-small). Recall that attention models have two types of layers, one small and large; and the Spectral norm (in particular, other norms do too) displays unusually small values for some of these layers for GPT. This Scale Collapse for the poorly trained GPT is similar to what we observed for the distilled ResNet20 model in Fig. 5b. Because of the anomalous Scale Collapse that is frequently observed in poorly trained models, these results suggest that scale-dependent norm metrics should not be directly applied to distinguish well-trained versus poorly trained models.

We next examine the distribution of $\alpha$ values in GPT versus GPT2-small. Figure 6b shows the histogram (empirical density), for all layers, of $\alpha$ for GPT and GPT2-small. The older deficient GPT has numerous unusually large $\alpha$ exponents—meaning they are not well-described by a PL fit. Indeed, we expect that a poorly trained model will lack good (i.e., small $\alpha$) PL behavior in many/ most layers. On the other hand, the newer improved GPT2-small model has, on average, smaller $\alpha$ values than the older GPT, with all $\alpha \leq 6$ and with smaller mean/median $\alpha$. It also has far fewer unusually large outlying $\alpha$ values than GPT. From this (and other results not shown), we see that $\bar{\alpha}$ from Eq. (12), provides a good quality metric for comparing the poorly trained GPT versus the well-trained GPT2-small. This should be contrasted with the behavior displayed by scale-dependent metrics such as the Frobenius norm (not shown) and the Spectral norm. This also reveals why $\hat{\alpha}$ performs unusually in Table 2. The PL exponent $\alpha$ behaves as expected, and thus the scale-invariant $\bar{\alpha}$ metric lets us identify potentially poorly trained models. It is the Scale Collapse that causes problems for $\hat{\alpha}$ (recall that the scale enters into $\hat{\alpha}$ via the weights $\log \lambda_{max}$).

Figure 7b plots $\alpha$ versus the depth (layer id) for each model. The deficient GPT model displays two trends in $\alpha$, one stable with $\alpha \sim 4$, and one increasing with layer id, with $\alpha$ reaching as high as 12. In contrast, the well-trained GPT2-small model shows consistent and stable patterns, again with one stable $\alpha \sim 3.5$ (and below the GPT trend), and the other only slightly trending up, with $\alpha \leq 6$. These results show that the behavior of $\alpha$ across layers differs significantly between GPT and GPT2-small, with the better GPT2-small looking more like the better ResNet-1K from Fig. 4b. These results also suggest that smaller more stable values of $\alpha$ across depth is beneficial, i.e., that the Correlation Flow is also a useful concept for NLP models.

**Table 2 Average value for the average Log Norm and Weighted Alpha metrics for pretrained OpenAI GPT and GPT2 models.**

| Series | # | $\langle \log \| \mathbf{W} \|_F \rangle$ | $\langle \log \| \mathbf{W} \|_\infty \rangle$ | $\hat{\alpha}$ | $\langle \log \| \mathbf{X} \|_\alpha^\alpha \rangle$ |
|---|---|---|---|---|---|
| GPT | 49 | 1.64 | 1.72 | 7.01 | 7.28 |
| GPT2-small | 49 | 2.04 | 2.54 | 9.62 | 9.87 |
| GPT2-medium | 98 | 2.08 | 2.58 | 9.74 | 10.01 |
| GPT2-large | 146 | 1.85 | 1.99 | 7.67 | 7.94 |
| GPT2-xl | 194 | 1.86 | 1.92 | 7.17 | 7.51 |

Column # refers to number of layers treated. Averages do not include the first embedding layer (s) because they are not (implicitly) normalized. GPT has 12 layers, with 4 Multi-head Attention Blocks, giving 48 layer Weight Matrices, **W**. Each Block has 2 components, the Self Attention (attn) and the Projection (proj) matrices. Self-attention matrices are larger, of dimension (2304 × 768) or (3072 × 768). The projection layer concatenates the self-attention results into a vector (of dimension 768). This gives 50 large matrices. Because GPT and GPT2 are trained on different data sets, the initial Embedding matrices differ in shape. GPT has an initial Token and Positional Embedding layers, of dimension (40478 × 768) and (512 × 768), respectively, whereas GPT2 has input Embeddings of shape (50257 × 768) and (1024 × 768), respectively. The OpenAI GPT2 (English) models are: GPT2-small, GPT2-medium, GPT2-large, and GPT2-xl, having 12, 24, 36, and 48 layers, respectively, with increasingly larger weight matrices.

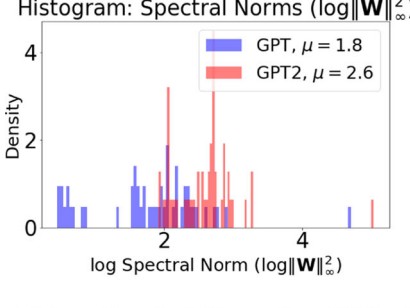 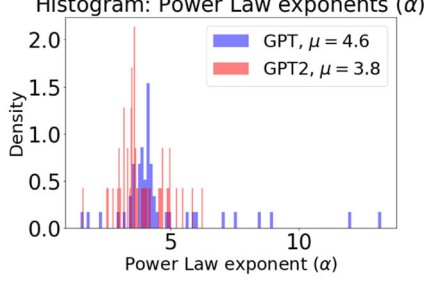

(a) Log Spectral Norm $(\log \| \mathbf{W} \|_\infty)$　　　　　(b) PL exponent $(\alpha)$

**Fig. 6 Histogram of PL exponents and Log Spectral Norms for NLP models.** Histogram of Log Spectral Norms (in **a**) and PL exponents (in **b**) for weight matrices from the OpenAI GPT and GPT2-small pretrained models.

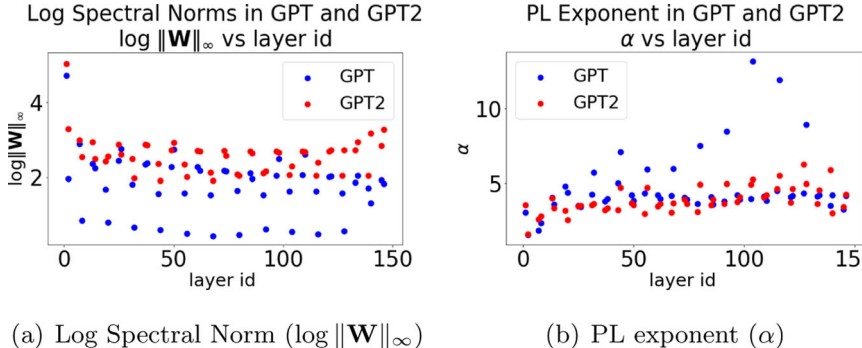

(a) Log Spectral Norm ($\log \|\mathbf{W}\|_\infty$)  (b) PL exponent ($\alpha$)

**Fig. 7 Log Spectral Norms and PL exponents for NLP models.** Log Spectral Norms (in (**a**)) and PL exponents (in (**b**)) for weight matrices from the OpenAI GPT and GPT2-small pretrained models. (Note that the quantities shown on each Y axis are different). In the text, this is interpreted in terms of Scale Collapse and Correlation Flow.

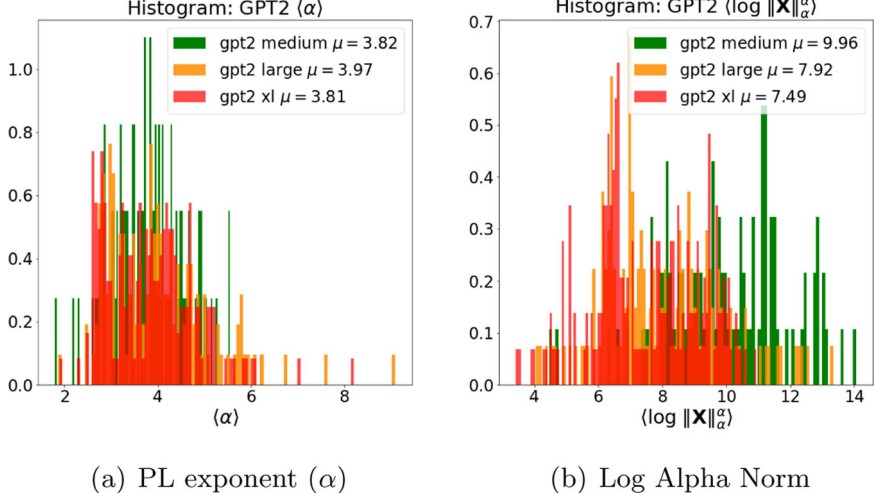

(a) PL exponent ($\alpha$)  (b) Log Alpha Norm

**Fig. 8 Histogram of PL exponents and Log Alpha Norm for weight matrices from models of different sizes in the GPT2 architecture series.** (Plots omit the first 2 (embedding) layers, because they are normalized differently giving anomalously large values).

We now look across series of increasingly improving GPT2 models (well-trained versus very-well-trained models), by examining both the PL exponent $\alpha$ as well as the Log Norm metrics. Figure 8 shows the histograms over the layer weight matrices for fitted PL exponent $\alpha$ and the Log Alpha Norm metric. In general, and as expected, as we move from GPT2-medium to GPT2-xl, histograms for both $\alpha$ exponents and the Log Norm metrics downshift from larger to smaller values.

From Fig. 8a, we see that $\bar{\alpha}$, the average $\alpha$ value, decreases with increasing model size (3.82 for GPT2-medium, 3.97 for GPT2-large, and 3.81 for GPT2-xl), although the differences are less noticeable between the differing well-trained versus very-well-trained GTP2 models than between the poorly trained versus well-trained GPT and GPT2-small models. Also, from Fig. 8b, we see that, unlike GPT, the layer Log Alpha Norms behave more as expected for GPT2 layers, with the larger models consistently having smaller norms (9.96 for GPT2-medium, 7.982 for GPT2-large, and 7.49 for GPT2-xl). Similarly, the Log Spectral Norm also decreases on average with the larger models (2.58 for GPT2-medium, 1.99 for GPT2-large, and 1.92 for GPT2-xl). As expected, the norm metrics can indeed distinguish among well-trained versus very-well-trained models.

While the means and peaks of the $\alpha$ distributions are getting smaller, towards 2.0, as expected, Fig. 8a also shows that the tails of the $\alpha$ distributions shift right, with larger GPT2 models having more unusually large $\alpha$ values. This is unexpected. It suggests that these larger GPT2 models are still under-optimized/over-parameterized (relative to the data on which they were trained) and that they have capacity to support datasets even larger than the recent XL 1.5B release[30]. This does not contradict recent theoretical work on the benefits of over-parameterization[31], e.g., since in practice these extremely large models are not fully optimized. Subsequent refinements to these models, and other models such as BERT, indicate that this is likely the case.

**Comparing hundreds of models**. We have performed a large-scale analysis of hundreds of publicly available models. This broader analysis is on a much larger set of CV and NLP models, with a more diverse set of architectures, that have been developed for a wider range of tasks; and it complements the previous more detailed analysis on CV and NLP models, where we have analyzed only a single architecture series at a time. See Supplementary Note 2 (and our publicly available repo) for details. To quantify the relationship between quality metrics and the reported test error and/or accuracy metrics, we use ordinary least squares to regress the metrics on the Top1 (and Top5) reported errors (as dependent variables), and we report the RMSE, the $R^2$ (R2) regresssion metric, and the Kendal-$\tau$ rank correlation metric. These include Top5 errors for the ImageNet-1K model, percent error for the CIFAR-10/100, SVHN, CUB-200-2011 models, and Pixel accuracy (Pix.Acc.) and Intersection-Over-Union (IOU) for

**Table 3 Comparison of linear regression fits for different average Log Norm and Weighted Alpha metrics across 5 CV datasets, 17 architectures, covering 108 (out of over 400) different pretrained DNNs.**

|  | $\log \| \cdot \|_F^2$ | $\log \| \cdot \|_\infty^2$ | $\hat{\alpha}$ | $\log \| \cdot \|_\alpha^\alpha$ |
|---|---|---|---|---|
| RMSE (mean) | 4.84 | 5.57 | 4.58 | 4.55 |
| RMSE (std) | 9.14 | 9.16 | 9.16 | 9.17 |
| R2 (mean) | 3.9 | 3.85 | 3.89 | 3.89 |
| R2 (std) | 9.34 | 9.36 | 9.34 | 9.34 |
| Kendal-tau (mean) | 3.84 | 3.77 | 3.86 | 3.85 |
| Kendal-tau (std) | 9.37 | 9.4 | 9.36 | 9.36 |

We include regressions only for architectures with five or more data points, and which are positively correlated with test error. These results can be readily reproduced using the Google Colab notebooks. (See Supplementary Note 2 for details.).

other models. We regress them individually on each of the norm-based and PL-based metrics.

Results are summarized in Table 3 (and Supplementary Note 2). For the mean, smaller RMSE, larger $R^2$, and larger Kendal-$\tau$ are desirable; and, for the standard deviation, smaller values are desirable. Taken as a whole, over the entire corpus of data, PL-based metrics are somewhat better for both the $R^2$ mean and standard deviation; and PL-based metrics are much better for RMSE mean and standard deviation. Model diagnostics (Supplementary Note 2) indicate many outliers and imperfect fits. Overall, though, these and other results suggest our conclusions hold much more generally.

## Discussion
Going beyond the goal of predicting trends in the quality of state-of-the-art neural networks without access to training or testing data, observations such as the layer-wise observations we described in Fig. 4 can be understood in terms of architectural differences between VGG, ResNet, and DenseNet. VGG resembles the traditional convolutional architectures, such as LeNet5, and consists of several [Conv2D-Maxpool-ReLu] blocks, followed by 3 large Fully Connected (FC) layers. ResNet greatly improved on VGG by replacing the large FC layers, shrinking the Conv2D blocks, and introducing residual connections. This optimized approach allows for greater accuracy with far fewer parameters, and ResNet models of up to 1000 layers have been trained[32].

The efficiency and effectiveness of ResNet seems to be reflected in the smaller and more stable $\alpha \sim 2.0$, across nearly all layers, indicating that the inner layers are very well correlated and more strongly optimized. This contrasts with the DenseNet models, which contains many connections between every layer. These results (large $\alpha$, meaning that even a PL model is probably a poor fit) suggest that DenseNet has too many connections, diluting high quality interactions across layers, and leaving many layers very poorly optimized. Fine-scale measurements such as these enable us to form hypotheses as to the inner workings of DNN models, opening the door to an improved understanding of why DNNs work, as well as how to design better DNN models. Correlation Flow and Scale Collapse are two such examples.

Statistical mechanics has long had influence on DNN theory and practice[33–35]. Our best-performing PL-based metrics are based on statistical mechanics via HT-SR Theory[1–3,34,36]. The way in which we (and HT-SR Theory) use statistical mechanics theory is quite different than the way it is more commonly formulated[33,35]. Going beyond idealized models, we use statistical mechanics in a broader sense, drawing upon techniques from quantitative finance, random matrix theory, and the statistical mechanics of heavy tailed and strongly correlated systems[34].

There is also a large body of work in ML on using norm-based metrics to bound generalization error[37–39]. This theoretical work aims to prove generalization bounds, and this applied work then uses these norms to construct regularizers to improve training. Proving generalization bounds and developing new regularizers is very different than our focus on validating pretrained models.

Our work also has intriguing similarities and differences with work on understanding DNNs with the information bottleneck principle[24,25], which posits that DNNs can be quantified by the mutual information between their layers and the input and output variables. Most importantly, our approach does not require access to any data, while information measures used in the information bottleneck approach do require this. Nevertheless, several results from HT-SR Theory, on which our metrics are based, have parallels in the information bottleneck approach. Perhaps most notably, the quick transition from a RANDOM-LIKE phase to BULK+SPIKES phase, followed by slow transition to a HEAVY-TAILED phase, as noted previously[1], is reminiscent of the dynamics on the Information Plane[25].

Finally, our work, starting in 2018 with the WeightWatcher tool[6], is the first to perform a detailed analysis of the weight matrices of DNNs[1–3]. Subsequent to the initial version of this paper, we became aware of two other works that were posted in 2020 within weeks of the initial version of this paper[40,41]. Both of these papers validate our basic result that one can gain substantial insight into model quality by examining weight matrices without access to any training or testing data. However, both consider smaller models drawn from a much narrower range of applications than we consider. Previous results in HT-SR Theory suggest that insights from these smaller models may not extend to the state-of-the-art CV and NLP models we consider.

We have developed and evaluated methods to predict trends in the quality of state-of-the-art neural networks—without access to training or testing data. Our main methodology involved weight matrix meta-analysis, using the publicly available Weight-Watcher tool[6], and informed by the recently developed HT-SR Theory[1–3]. Prior to our work, it was not even obvious that norm-based metrics would perform well to predict trends in quality across models (as they are usually used within a given model or parameterized model class, e.g., to bound generalization error or to construct regularizers). Our results are the first to demonstrate that they can be used for this important practical problem. Our results also demonstrate that PL-based metrics perform better than norm-based metrics. This should not be surprising—at least to those familiar with the statistical mechanics of heavy tailed and strongly correlated systems[8,21–23]—since our use of PL exponents is designed to capture the idea that well-trained models capture information correlations over many size scales in the data. Again, though, our results are the first to demonstrate this. Our approach can also be used to provide fine-scale insight (rationalizing the flow of correlations or the collapse of size scale) throughout a network. Both Correlation Flow and Scale Collapse are important for improved diagnostics on pretrained models as well as for improved training methodologies.

More generally, our results suggest what a practical theory of DNNs should look like. To see this, let's distinguish between two types of theories: non-empirical or analogical theories, in which one creates, often from general principles, a very simple toy model that can be analyzed rigorously, and one then claims that the model is relevant to the system of interest; and semi-empirical theories, in which there exists a rigorous asymptotic theory, which comes with parameters, for the system of interest, and one then adjusts or fits those parameters to the finite non-asymptotic data, to make predictions about practical problems. A drawback of the former approach is that it typically makes very strong assumptions, and the strength of those assumptions can limit the

practical applicability of the theory. Nearly all of the work on DNN theory focuses on the former type of theory. Our approach focuses on the latter type of theory. Our results, which are based on using sophisticated statistical mechanics theory and solving important practical DNN problems, suggests that the latter approach should be of interest more generally for those interested in developing a practical DNN theory.

## Methods

To be fully reproducible, we only examine publicly available, pretrained models. All of our computations were performed with the `WeightWatcher` tool (version 0.2.7)[6], and we provide all Jupyter and Google Colab notebooks used in an accompanying github repository[7], which includes more details and more results.

**Additional details on layer weight matrices**. Recall that we can express the objective/optimization function for a typical DNN with $L$ layers and with $N \times M$ weight matrices $\mathbf{W}_l$ and bias vectors $\mathbf{b}_l$ as Eq. (2). We expect that most well-trained, production-quality models will employ one or more forms of regularization, such as Batch Normalization (BN), Dropout, etc., and many will also contain additional structure such as Skip Connections, etc. Here, we will ignore these details, and will focus only on the pretrained layer weight matrices $\mathbf{W}_l$. Typically, this model would be trained on some labeled data $\{d_i, y_i\} \in \mathcal{D}$, using Backprop, by minimizing the loss $\mathcal{L}$. For simplicity, we do not indicate the structural details of the layers (e.g., Dense or not, Convolutions or not, Residual/Skip Connections, etc.). Each layer is defined by one or more layer 2D weight matrices $\mathbf{W}_l$, and/or the 2D feature maps $\mathbf{W}_{l,i}$ extracted from 2D Convolutional (Conv2D) layers. A typical modern DNN may have anywhere between 5 and 5000 2D layer matrices.

For each Linear Layer, we get a single $(N \times M)$ (real-valued) 2D weight matrix, denoted $\mathbf{W}_l$, for layer $l$. This includes Dense or Fully Connected (FC) layers, as well as 1D Convolutional (Conv1D) layers, Attention matrices, etc. We ignore the bias terms $\mathbf{b}_l$ in this analysis. Let the aspect ratio be $Q = \frac{N}{M}$, with $Q \geq 1$. For the Conv2D layers, we have a 4-index Tensor, of the form $(N \times M \times c \times d)$, consisting of $c \times d$ 2D feature maps of shape $(N \times M)$. We extract $n_l = c \times d$ 2D weight matrices $\mathbf{W}_{l,i}$, one for each feature map $i = [1, \ldots, n_l]$ for layer $l$.

**SVD of convolutional 2D layers**. There is some ambiguity in performing spectral analysis on Conv2D layers. Each layer is a 4-index tensor of dimension $(w, h, in, out)$, with an $(w \times h)$ filter (or kernel) and $(in, out)$ channels. When $w = h = k$, it gives $(k \times k)$ tensor slices, or pre-Activation Maps, $\mathbf{W}_{i,L}$ of dimension $(in \times out)$ each. We identify 3 different approaches for running SVD on a Conv2D layer:

1. run SVD on each pre-Activation Map $\mathbf{W}_{i,L}$, yielding $(k \times k)$ sets of $M$ singular values;
2. stack the maps into a single matrix of, say, dimension $((k \times k \times out) \times in)$, and run SVD to get $in$ singular values;
3. compute the 2D Fourier Transform (FFT) for each of the $(in, out)$ pairs, and run SVD on the Fourier coefficients[42], leading to $\sim (k \times in \times out)$ non-zero singular values.

Each method has tradeoffs. Method (3) is mathematically sound, but computationally expensive. Method (2) is ambiguous. For our analysis, because we need thousands of runs, we select method (1), which is the fastest (and is easiest to reproduce).

**Normalization of empirical matrices**. Normalization is an important, if under-appreciated, practical issue. Importantly, the normalization of weight matrices does *not* affect the PL fits because $\alpha$ is scale-invariant. Norm-based metrics, however, do depend strongly on the scale of the weight matrix—that is the point. To apply RMT, we usually define $\mathbf{X}$ with a $1/N$ normalization, assuming variance of $\sigma^2 = 1.0$. Pretrained DNNs are typically initialized with random weight matrices $\mathbf{W}_0$, with $\sigma^2 \sim 1/\sqrt{N}$, or some variant, e.g., the Glorot/Xavier normalization[43], or a $\sqrt{2/Nk^2}$ normalization for Convolutional 2D Layers. With this implicit scale, we do *not* "renormalize" the empirical weight matrices, i.e., we use them as-is. The only exception is that *we do rescale* the Conv2D pre-activation maps $\mathbf{W}_{i,L}$ by $k/\sqrt{2}$ so that they are on the same scale as the Linear/Fully Connected (FC) layers.

**Special consideration for NLP models**. NLP models, and other models with large initial embeddings, require special care because the embedding layers frequently lack the implicit $1/\sqrt{N}$ normalization present in other layers. For example, in GPT, for most layers, the maximum eigenvalue $\lambda_{max} \sim \mathcal{O}(10 - 100)$, but in the first embedding layer, the maximum eigenvalue is of order $N$ (the number of words in the embedding), or $\lambda_{max} \sim \mathcal{O}(10^5)$. For GPT and GPT2, we treat all layers as-is (although one may want to normalize the first 2 layers $\mathbf{X}$ by $1/N$, or to treat them as outliers).

## Data availability

Data analyzed during the study are all publicly available; and data generated during the study are available along with the code to generate them in our public repository (https://github.com/CalculatedContent/ww-trends-2020).

## Code availability

Code sufficient to generate the results of the study is available in our public repository (https://github.com/CalculatedContent/ww-trends-2020).

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

## Acknowledgements

M.W.M. would like to acknowledge ARO, DARPA, NSF, and ONR as well as the UC Berkeley BDD project and a gift from Intel for providing partial support of this work. Our conclusions do not necessarily reflect the position or the policy of our sponsors, and no official endorsement should be inferred. We would also like to thank Amir Khos-rowshahi and colleagues at Intel for helpful discussion regarding the Group Regularization distillation technique.

## Author contributions

CHM and MWM designed the research, performed the research, and wrote the paper. TSP performed the research and edited the paper.

## Competing interests

The authors declare no competing interests.
