## [Peer Review File · Nature Communications]

Reviewer #1 (Remarks to the Author):

The paper presents a set of metrics to evaluate the quality of trained neural networks without having access to the training/testing data used to train/validate the models. This is achieved by taking advantage of regularization techniques to train neural networks. The paper evaluates different approaches such as norms of the learned weight matrices or parameters from power law fits of the eigenvalues of weight matrices. These proposed metrics are therefore evaluated on a large number of pre-trained convolutional neural network architectures and natural language processing models.

I like the idea of proposing a metric to evaluate the quality of already trained models. However, I believe the paper arrives to conclusions without using proper statistical analysis and it is too speculative:

- The average quality vs Top 1 accuracy is tested using linear regression. However, the p-values are not reported and only the RMSE and R^2 is provided. Moreover, the assumption of normality of residuals is not tested. Therefore, we cannot conclude with RMSE and R^2 that there is a correlation between the metrics and the Top 1 accuracy.
- In the layer analysis, the paper analyzes how the PL exponent varies in the different layers. Figure 4 shows how the distribution of alpha among layers is architecture-dependent, having different distributions. Computing the mean of this value among all layers is in my opinion not a good metric to compare different models. Despite the fact, that the results of the linear regression used in Figure 9 are not reported, it seems to be the case as seen in the figure, that there is a poor correlation between trained models with different architectures.
- In the first paragraph of the NLP models (line 325) the paper indicates that lower quality metrics represent better model, but Table 2 indicates the opposite. Also, when comparing the metrics with the depth of the network the paper compares GPT2-medium, large, and xl. However, it does not include in the comparison the small model, which will not present the nice trend of decreasing values of the metrics when increasing the depth of the model.
- The paper also concludes that the alpha metric is a good metric to explain the quality based on Figure 6 and 8. However, these histograms look very similar with the exception of a few outliers. This should be analyzed in more detail.
- The paper refers to results to support the conclusion, but these results are not reported in the paper or supplementary material -- "not shown here". If these results support the conclusion, they should be presented in the paper or at least the supplementary material.

So, while I like the idea and believe that the authors tackle an important topic, I believe that the conclusions they arrive at, are not necessarily supported by the experiments and the shown results. Hence, I believe the manuscript needs a major revision and additional experiments in order to support the claims. For these reasons my recommendation is to reject the paper, and encourage the authors to proceed with their work.

Reviewer #3 (Remarks to the Author):

Nowadays, much research effort is being put into developing a theory for deep learning that could predict a network's generalization error. Yet, for many of these theories, the rubber never meets the road, as no concrete guidelines are provided for practitioners that would allow them to predict the generalization error of state-of-the-art networks, trained on real data, across various hyperparameters, at scale. The authors of this paper undertake this important challenge in the specific, yet prevalent setting of transfer learning (and also model distillation), proposing novel statistics for predicting generalization performance, based on the spectrum of layer-wise weight matrices. Of particular interest is the methodology that the authors chose, relying heavily on data scientific experiments. This work exemplifies the potency of data science in delivering knowledge

in the modern scientific era. I, therefore, recommend the acceptance of this publication to Nature Communications, once the authors address the minor issues raised below.

- "DNNs will exhibit correlations over many size scales" -- could the authors elaborate on this statement and explain it for readers unaware of theories pertaining to strongly-correlated systems?
- Would the authors consider the following logic to be correct:
 - Smaller power-law exponents result in a more condensed spectrum;
 - More condensed spectrum results in a smaller condition number;
 - A smaller condition number facilitates the propagation of information/features across layers.
- Could the authors elucidate the relation between the Information Bottleneck and the Theory of Heavy-Tailed Self Regularization?
- Would the authors consider rank collapse a phenomenon detrimental or beneficial to the network's performance? Why?
- Could the authors define mathematically what is "Scale Collapse"?
- If one were to multiply all weights in the network by a constant, the spectral norm would change, but the classification error would not. Isn't the spectral norm of the weights therefore irrelevant for diagnosing deep networks?
- The authors mention in the paper that weight matrices can be "over-parameterized, relative to the amount of data". Is there any empirical evidence in the literature showing that networks can have worse generalization performance due to overparameterization? Isn't the double descent phenomenon showing that generalization error keeps decreasing after the second descent despite increasing overparameterization?
- Not all losses used in deep learning are a function of the difference between the network's prediction and the label (for example, cross-entropy). I would suggest changing the loss in Equation 1 to include two inputs: the prediction and the label.

Vardan Papyan

Response to Reviewers “Predicting trends in the quality of state-of-the-art neural networks without access to training or testing data”

Thanks to the two reviewers for providing feedback. One reviewer had questions about additional results to justify our main claims. In some cases, we had these results previously, but they were not included due to space constraints; and, in other cases, these results were easy to generate. To do so, we have completely redone our analysis, including the requested analysis as well as additional analysis, and we are happy to include these additional results. This reviewer also had several misunderstandings, which we will clarify. The other reviewer had several general questions about how our conclusions fit into the bigger picture that is emerging in the area, and we are also happy to address and clarify these issues. We go through each of the points in turn below, with the reviewer comments in italics and our response immediately after.

Reviewer 1

The paper presents a set of metrics to evaluate the quality of trained neural networks without having access to the training/testing data used to train/validate the models. This is achieved by taking advantage of regularization techniques to train neural networks. The paper evaluates different approaches such as norms of the learned weight matrices or parameters from power law fits of the eigenvalues of weight matrices. These proposed metrics are therefore evaluated on a large number of pre-trained convolutional neural network architectures and natural language processing models.

To clarify a potential misunderstanding, it is not correct that the metrics we consider are “achieved by taking advantage of regularization techniques to train neural networks.” Norms may be used to train neural networks, but power law fits are not. Instead, for the power law fits, we use ideas from statistical mechanics and strongly correlated system theory to try to capture the idea that trained neural networks encode correlations from the data. It is fair to say that the power law exponents characterize is the implicit regularization, whether or not the model trainer knew that that was what he or she was doing, but not any regularization explicitly used to train. We have clarified this point in the revised manuscript.

I like the idea of proposing a metric to evaluate the quality of already trained models. However, I believe the paper arrives to conclusions without using proper statistical analysis and it is too speculative:

- The average quality vs Top 1 accuracy is tested using linear regression. However, the p -values are not reported and only the RMSE and R^2 is provided. Moreover, the assumption of normality of residuals is not tested. Therefore, we cannot conclude with RMSE and R^2 that there is a correlation between the metrics and the Top 1 accuracy.

The reviewer is correct. We intended for the reader to examine plots to see that the trends are qualitatively in the correct (or incorrect, in some cases for norm-based metrics) direction, and then to look at the numbers to get quantitative insight. We know that errors are not normal; so, for this and other reasons, in our experience, reporting p -values won’t clarify the situation. Instead, to clarify understanding, we have expanded all of the tables that report RMSE and/or R^2 to report the Kendal- τ rank correlation measure; and we have provided figures in the Supplementary Information

for all model-dataset pairs, which indicate trends more clearly than the aggregated data, and which (among other things) show that the plots display the 95% confidence band for every regression along with tabulated results for all metrics. Also, a point we didn't emphasize enough is that from HT-SR theory we expect that PL-based metrics to have strong linear and not just rank correlation. We have clarified these points in the text of the revised manuscript, and in the tables and plots throughout the manuscript. (Also, additional results beyond what we have in the main text and Supplementary Information are available in our fully-reproducible and publicly-available repo, and it is our hope that that level of transparency will aid the reader's understanding.)

- *In the layer analysis, the paper analyzes how the PL exponent varies in the different layers. Figure 4 shows how the distribution of alpha among layers is architecture-dependent, having different distributions. Computing the mean of this value among all layers is in my opinion not a good metric to compare different models. Despite the fact, that the results of the linear regression used in Figure 9 are not reported, it seems to be the case as seen in the figure, that there is a poor correlation between trained models with different architectures.*

The reviewer is correct that the distribution of α is different for different architectures. That is the point. Different architectures can be more or less appropriate for a given task, and within an architecture series the metrics we present can be used to predict trends in model quality. We never claim that there is a correlation between different architectures, and we know that there is not. There are many reasons for this, e.g., different normalizations, etc. We have clarified this point in the revised manuscript.

As for (previous) Figure 9, this should be read in light of Tables 7, 8, and 9, as well as the discussion in Sec 2.4. Our metrics for predicting trends in model quality perform well within a given architectural series, and we have discussed this in detail in the main text. We also want to show that our general results hold for the hundreds of publicly-available CV and NLP models. All of this is available in our publicly-available repo, and so we decided in the text and Supplementary Information to go with just a summary. In the main text, we have R2, MSE, and Kendall-Tau; and in the appendix, we have more granular results for several model series. Since (previous) Figure 9 partitions based on data and not model series, we think that it will likely confuse the reader, as opposed to presenting results for all model-dataset pairs, as we now do. Thus, we removed it, and we will simply point to the publicly-available repo. We have clarified these points in the revised manuscript.

- *In the first paragraph of the NLP models (line 325) the paper indicates that lower quality metrics represent better model, but Table 2 indicates the opposite. Also, when comparing the metrics with the depth of the network the paper compares GPT2-medium, large, and xl. However, it does not include in the comparison the small model, which will not present the nice trend of decreasing values of the metrics when increasing the depth of the model.*

Sorry for the confusion. We have adjusted that table, and we have substantially reworded the text.

The first point is there are two things we are illustrating: one is when we increase both data size and model size (GPT2-small to GPT2-medium to GPT2-large to GPT2-xl); and the second is when a given model is trained to high-quality versus low-quality data (GPT2-small versus GPT). For the former, as the model size increases, we see

a general trend with all the metrics decreasing, with the largest difference between GPT2-medium and GPT2-large, and with the difference between GPT2-small and GPT2-medium being very small (near the noise level, and slightly in the opposite direction). For the latter, when comparing the well trained to poorly trained model, we see that the smaller and poorly trained GPT model seems to violate what we expect. Thus, we look in more detail at the two contributions to $\hat{\alpha}$, the PL exponent, α , and the spectral norm, $\|\cdot\|_\infty$. Doing this makes it clear what while the α part behaves as HT-SR theory would predict, i.e., that with better models the PL exponents get smaller, the norms behave anomalously and exhibit Scale Collapse. The bottom line is that the metrics we discuss can be used to distinguish between well-trained versus very well trained models, but that one has to look at both the PL exponents and Spectral Norm to diagnose the model problems; this was one of our original points. We have clarified these points in the revised manuscript.

- *The paper also concludes that the alpha metric is a good metric to explain the quality based on Figure 6 and 8. However, these histograms look very similar with the exception of a few outliers. This should be analyzed in more detail.*

The details matter a lot in both of these figures. The reviewer is correct that the plots are similar—the important point is how they are different. In Fig 6a, the alphas behave as they should and as as would be predicted by HT-SR theory, i.e., the better GPT2 has the bulk of the alphas shifted to the left and all of the large outlying alphas removed. In Fig 6b, the spectral norm fails to perform well; while it is true that the higher-quality GPT2 does not exhibit Scale Collapes, the distribution of norms is shifted slightly to the right. In Fig 8a, there is a slight shift of alphas to the left as the models get larger (with average values now given in the text). Interesting the larger models still have some larger alpha values, and we conjecture that the reason for this is that the models are not fully optimized. Subsequent developments in NLP have shown this to be the case. In Fig 8b, which shows the trend more strongly, the Shatten norm shifts to the left with higher quality models, as we would expect. We have put this discussion and clarified these points in the text.

- *The paper refers to results to support the conclusion, but these results are not reported in the paper or supplementary material – “not shown here”. If these results support the conclusion, they should be presented in the paper or at least the supplementary material.*

We have removed those statements and only refer to the main text or the Supplementary Information. We do point out, however, that we intend our results to be fully-reproducible by the community, and thus our publicly-available repo is designed to contain the enormous quantity of data underlying our analysis.

So, while I like the idea and believe that the authors tackle an important topic, I believe that the conclusions they arrive at, are not necessarily supported by the experiments and the shown results. Hence, I believe the manuscript needs a major revision and additional experiments in order to support the claims. For these reasons my recommendation is to reject the paper, and encourage the authors to proceed with their work.

Thanks for the comments. Space is limited in the main text, but hopefully we have addressed your main concerns.

Reviewer 3

Nowadays, much research effort is being put into developing a theory for deep learning that could predict a network's generalization error. Yet, for many of these theories, the rubber never meets the road, as no concrete guidelines are provided for practitioners that would allow them to predict the generalization error of state-of-the-art networks, trained on real data, across various hyperparameters, at scale. The authors of this paper undertake this important challenge in the specific, yet prevalent setting of transfer learning (and also model distillation), proposing novel statistics for predicting generalization performance, based on the spectrum of layer-wise weight matrices. Of particular interest is the methodology that the authors chose, relying heavily on data scientific experiments. This work exemplifies the potency of data science in delivering knowledge in the modern scientific era. I, therefore, recommend the acceptance of this publication to *Nature Communications*, once the authors address the minor issues raised below.

Thanks for articulating at least as well as we did the broad goal of the paper.

- "DNNs will exhibit correlations over many size scales" – could the authors elaborate on this statement and explain it for readers unaware of theories pertaining to strongly-correlated systems?

We have elaborated on this in the revised manuscript.

- Would the authors consider the following logic to be correct:
- Smaller power-law exponents result in a more condensed spectrum;
- More condensed spectrum results in a smaller condition number;
- A smaller condition number facilitates the propagation of information/features across layers.

More important than the smaller condition number, which has to do with the ratio of the largest to smallest eigenvalue, is the "shape" of the spectral distribution, and in particular the details of the tail and how the tail is estimated. We have clarified this point in the revised manuscript.

- Could the authors elucidate the relation between the Information Bottleneck and the Theory of Heavy-Tailed Self Regularization?

Certainly. We have clarified this point in the revised manuscript.

- Would the authors consider rank collapse a phenomenon detrimental or beneficial to the network's performance? Why?

In general, it would be detrimental. This is known in GANs, where regularizers are used to prevent rank collapse, but it appears to hold more generally. We have clarified this point in the revised manuscript.

- Could the authors define mathematically what is "Scale Collapse"?

By Scale Collapse, we mean that the size scale or norm of one or more layers changes a lot, while the size scale or norm of the remaining layers changes very little, as a function of some change to or perturbation of a model. Examples of such changes could include model compression or distillation, data augmentation, additional training, model fine-tuning, etc. A more precise definition would have to do with the stability of the model under those sorts of perturbations, but we think that including such a precise definition in the main text would only confuse the reader. We have clarified this point in the revised manuscript.

- *If one were to multiply all weights in the network by a constant, the spectral norm would change, but the classification error would not. Isn't the spectral norm of the weights therefore irrelevant for diagnosing deep networks?*

Yes and No. In theory, one could do that; but in practice, for state-of-the-art models, one does not do that. Recall we are comparing networks with similar architectures. One then has to choose (explicitly, or more commonly implicitly, by design choices in the training of the model) a canonical way to normalize the network so that similar architectures can be compared. While different networks do have different normalizations, we have not found any discrepancies across models in the same architecture series. We have clarified this point in the revised manuscript.

- *The authors mention in the paper that weight matrices can be “over-parameterized, relative to the amount of data.” Is there any empirical evidence in the literature showing that networks can have worse generalization performance due to overparameterization? Isn't the double descent phenomenon showing that generalization error keeps decreasing after the second descent despite increasing overparameterization?*

Superficially, work on double descent may be interpreted as “more parameters are better,” but in practice the situation is not that simple, e.g., since it can be difficult to define an appropriate effective number of parameters in realistic architectures and since realistic models are not fully optimized. We comment on and clarify this point in the revised manuscript.

- *Not all losses used in deep learning are a function of the difference between the network's prediction and the label (for example, cross-entropy). I would suggest changing the loss in Equation 1 to include two inputs: the prediction and the label.*

That is a good point. We have modified this equation and adjusted the surrounding and subsequent text.

Reviewer #1 (Remarks to the Author):

First, I would like to thank the authors for taking my comments into account when revising the submission. The authors did a good job incorporating additional experiments to support their claims, addressing my concerns, and clarifying my misunderstandings. All this was done in the short period of time of this revision process, for which I congratulate the authors. In particular, I believe that performing a more robust statistical analysis of the results greatly improves the submission. For this reason, I would like to change my recommendation and suggest to accept the article.

Reviewer #3 (Remarks to the Author):

The authors have addressed my comments. I, therefore, recommend the acceptance of this publication to Nature Communications.